# Process-Driven Layout Optimization of a Portable Hybrid Manufacturing Robotic Cell Structure

Harry Bikas, Dimitrios Manitaras, Thanassis Souflas and Panagiotis Stavropoulos *

Laboratory for Manufacturing Systems & Automation, University of Patras, 26504 Patras, Greece;
bikas@lms.mech.upatras.gr (H.B.); manitaras@lms.mech.upatras.gr (D.M.); souflas@lms.mech.upatras.gr (T.S.)
* Correspondence: pstavr@lms.mech.upatras.gr; Tel.: +30-2610-910160

**Abstract:** Hybrid manufacturing combines manufacturing processes (typically additive manufacturing and machining) exploiting the benefits of each and enabling repair scenarios. Such an approach can be integrated with a robot, and if made portable, can form a flexible machine tool that can be easily transported anywhere to enable repairs in the field. However, the design of the load-bearing structure determines its transportability, and its stiffness plays a crucial functional role under dynamic loads and affects the product quality. Finding the right balance between weight and stiffness requires accurate boundary conditions and an effective design. In this work, a method is proposed towards process-driven optimization of a portable manufacturing cell structure. The dynamic cutting forces are determined through a machining process model and, via a simplified model of the robot arm, the forces at the base of the robot are estimated. Since the frame consists of beams, the layout optimization method is applied, using the estimated process forces as boundary conditions to optimize the arrangement of beams. The proposed method achieved 0.05 mm displacement in the load-bearing structure under milling and an acceptable accuracy of the position of a hole's center during drilling, while the overall weight reduced by 17.6%.

**Keywords:** hybrid manufacturing; machining process simulation; layout optimization; machine frame design; dynamic response analysis

## 1. Introduction

Hybrid manufacturing (HM) is one of the manufacturing processes that has been developed to meet the industry's specific requirements as it constitutes a versatile and efficient manufacturing method. Combining high-value additive manufacturing with precision subtractive processes such as milling and grinding in a single machine tool is a logical step in the evolution of manufacturing processes and is known as HM [1]. Rather than encouraging waste, HM can support the remanufacturing and repair of high-value components. By adding material only where it is needed, it reduces material waste and can extend the life of components and equipment [2].

Industrial robots' use has been constantly expanding in the manufacturing sector due to their unique advantages. Robotic machining is still a new application for robots, and until now, their industrial adoption is restricted to tasks with low cutting force requirements (e.g., drilling and polishing). However, their use for more challenging tasks, such as milling, is driven by the multiple advantages they present over conventional CNC machines [3,4]. Robotic arms offer a high degree of flexibility due to their kinematics, and a large working envelope with minimal work floor footprint, which justifies their use in numerous manufacturing processes. They can be considered more attractive than machine tools with several degrees of freedom (such as five-axis machine tools) because they can be integrated into different industrial environments and combine different processes in a single machine [5]. In addition, the combination of a robotic arm with HM is a flexible option and has many advantages [6].

The repair capability and the manufacturing of complex parts via HM can be further extended if the robotic cell is portable [7]. A portable robotic repair cell can open new opportunities for in situ repair of high-value parts with long lead times for various industries, especially in applications with repeated failures. This approach can shorten the supply chain of a production or maintenance operation by enabling supply management practices such as just-in-time manufacturing. Waste is reduced in terms of time and inventory, as available stock can be significantly reduced [8].

Portable robotic hybrid manufacturing units are a promising solution for decentralized (re)manufacturing. However, the design of such portable robotic cells is challenging since the main structure of the cell must be both lightweight to improve portability and stiff enough to withstand the loads imposed by the manufacturing process (particularly machining). Such cells often need to be customized to accommodate different part sizes and shapes, which dictates their overall dimensions, as well as the forces involved in the manufacturing processes.

Hybrid manufacturing systems integrated in robotic arms are more complicated and less robust than a dedicated machine tool. Due to these factors, full hybrid systems built on portable robotic platforms with several degrees of freedom are not yet commercially available off-the-shelf. However, companies such as PROMATION can create bespoke cells for a range of industries and have investigated HM solutions based on robotic arms [9]. Other systems include more than one robot working together (or even in combination with a machining system) to produce large-scale parts. Typically, each subsystem is responsible for only one process and the whole system achieves the desired requirements [10,11]. Besides market-available solutions, robotic systems integrated with HM have been reported for research or educational purposes [12,13].

On the other hand, there are some examples of industrial robot machining cases. HUST-Wuxi Research Institute developed and constructed a robotic milling system to process large marine propellers [14]. The Fanuc M-710iC/12 robot is robust and precise, making it a go-to for intricate milling tasks in materials ranging from metals to composites. Another example is the ABB IRB 2600 robot, which offers outstanding versatility and agility, ideal for precision milling in automotive and aerospace components. These robots are part of larger robotic cells that are designed to automate and streamline the milling process, enhancing efficiency and precision in industrial manufacturing settings [15].

The stiffness of a machine tool has a significantly larger impact on the quality of the machining process since the forces in mechanical material removal processes far exceed those in AM processes (which are negligible and mostly due to the inertia of the process head) [16]. Numerous attempts have been made in the literature, particularly for very precise manufacturing processes, to close the loop between the forces generated by the process and the design of the machine tool structure. A dynamic framework that combines design and modeling has been employed by researchers to construct a micro-milling machine tool. They were able to assess how their design affected process accuracy and account for it in their design optimization [17]. Within the same range, quality, and economic efficiency KPIs were integrated throughout the machine tool's design phase to recommend the best option for each application [18]. Another concept tool based on Finite Element modeling has been developed [19]. This tool can be utilized in the design phase and includes kinematics, multi-body simulations, and dynamic modeling of processes. A different framework, built on portable platforms, has been created specifically for machining. It aids in the design of these portable machine tools and forecasts the behavior of the entire system [20]. Aside from research related to portable robotic cells, the industry is also working to develop solutions that may be applied to difficult or distant situations [21].

This paper proposes a structured approach to the optimization of a load-bearing structure for a portable robotic cell which is based on the forces generated by the manufacturing processes involved. A fast-running simulation model is used to predict the stiffness of

the robotic arm, while process-generated forces are also used towards running a layout optimization algorithm to optimize the design of the cell structure.

## 2. Materials and Methods

The overall approach to the design of the cell frame is based on the concept of integrated machining process modeling. To provide accurate boundary conditions for the dynamic analysis and optimization of the robot cell support structure, knowledge of the input loading conditions is required. These loading requirements are influenced by both the dynamic behavior of the robot during operation (inertial loads, pose-dependent stiffness) and the physical mechanism of the manufacturing process itself.

An overview of the overall strategy is provided in Figure 1. The process starts with the definition of the manufacturing processes involved and desired working envelope, followed by the definition of a basic cell architecture and layout. A conceptual design is created to form the starting point for the optimization algorithm. In parallel, based on the envisioned application scenario and using the basic layout of the cell, a CAM tool is utilized to generate toolpaths and establish process parameters. These are subsequently used to calculate the involved process forces, which then propagate through the robot structure to the frame. These are used as boundary conditions for the layout optimization algorithm. The results of the layout optimization are post-processed to aid manufacturability, and a final dynamic simulation is carried out to validate results. Each of the main technological building blocks is described in detail in the following sections.

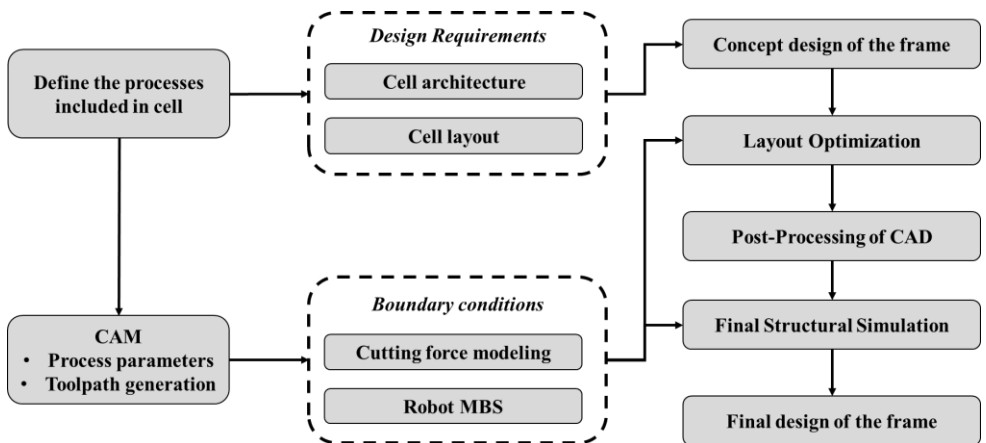

**Figure 1.** Proposed framework for the design optimization of the cell frame.

### 2.1. Machining Process Simulation

An established mechanistic cutting force model is used to estimate the cutting forces based on the tool properties, workpiece material, and process parameters [22]. The existing literature can be used to determine the proper model coefficients for the combination of the selected cutting tool and material. The mechanistic model assumes that the cutting forces are a function of the immersion angle of the $j$-th cutting edge of the cutting tool ($\varphi j$) and are linearly related to the feed per tooth ($fz$) and the axial depth of cut ($z$). The cutting force coefficients ($Kc$) and edge force coefficients ($Ke$) can be obtained from the literature or measured experimentally. The instantaneous cutting forces in the tangential ($dFt,j$), radial ($dFr,j$) and axial ($dFa,j$) direction are integrated along the axial depth of cut ($z$) to calculate the cutting forces at each cutting edge over one revolution of the tool. The formulation of the cutting force model is shown in the following set of equations.

$$dFt,j \ (\varphi j, z) = KtedS(z) + Ktcfz \ \sin(\varphi j) \ dz$$

$$dFr,j \ (\varphi j, z) = KredS(z) + Krcfz \ \sin(\varphi j)dz$$

$$dFa,j\ (\varphi j, z) = KaedS(z) + Kacfz\ \sin(\varphi j)\ dz \qquad (1)$$

*2.2. Robot Multi-Body Simulation (MBS)*

In general, the low stiffness that robotic arms provide in comparison to their many degrees of freedom is one of their main drawbacks. As robotics are further developed, there are robotic arms with high structural stiffness which are used in the manufacturing processes and especially in machining [15]. However, the majority of robots are made up of open, serial kinematic chains with rotating joints to sustain them [23]. Pose-dependent stiffness results from a change in the mechanical compliance of the joints and links when a robot moves through various postures (positions and orientations). As a result, their dynamic behavior during machining is still inferior to that of machine tools, due to their vast differences in terms of design. Moreover, robot behavior can be impacted by gravity. As the posture varies, the orientation of each link is altered; so, joints can withstand variations in loads resulting from the weight of the robot and any attached tools. During the operation of the robot, this issue is solved because the robot controller takes into consideration the gravity forces during the calculation of the inverse kinematics and the determination of the required torque to be applied in the joint motors to achieve the desired motion. Furthermore, every time the inverse kinematics problem is handled, inaccuracies in joint locations are introduced. The cutting forces acting to the robot's edge during milling operations might result in notable deviations. Due to the non-constant stiffness caused by all the above, estimating the reaction forces as they vary dynamically requires the use of a computational model. Multi-body simulation is a useful technique for modeling robot behavior since it treats all of the robot's joints and links as elastic components. Also, the gravity effect is included in the estimation of the forces at the mount of the robot, as the inertial matrix of the robot is generated and included in the model. This model is based on the dynamics of the robot and uses the cutting forces that were computed in the previous section as input. The reaction forces at the mounting location of the robot are estimated using this model, and the results are used as input for the design of the load-bearing structure. Reaction forces can be calculated for the chosen process parameters in conjunction with the appropriate toolpath. In addition to displaying the deformation at the robot's mount, this simulation also estimates the forces at each location [24]. The development of the multi-body robot simulation is not a direct part of the work; however, it is used as an input to the overall approach.

*2.3. Layout Optimization of the Frame*

The next step of the proposed method involves optimizing the load-bearing structure of the cell. This can be modeled as a truss structure composed of welded steel beams. Towards optimizing the design, a layout optimization algorithm is deployed to determine the most effective layout of individual truss members for a given design area, set of loads, supports, and material attributes. A typical layout optimization process is made up of four steps:

1.  A design domain, load scenarios, and support conditions are provided.
2.  Nodes are used to discretize this design domain.
3.  Connecting nodes create potential members by creating a "ground structure".
4.  Finally, a linear programming problem is solved to determine the most effective member arrangement, which will be a subset of the "ground structure".

The LP formulation has been used for layout optimization for decades and, in recent years, it has been applied to multiple load case problems. The objective function is as follows:

$$\min V = I^T a \qquad (2)$$

Subject to:

$$for\ all\ a \in \mathbb{F}\ \{\ Bq^{\infty} = f^{\infty}$$
$$-\sigma - \alpha \leq q^{a} \leq \sigma + \alpha\ \}$$
$$\alpha \geq 0$$

where $V$ is the structural volume, and $l$ and $a$ are vectors of member lengths and areas, correspondingly. $B$ is an equilibrium matrix containing the direction cosines; $q$ is a vector containing the internal member forces and $f$ is a vector containing the external forces. Also, $\sigma+$ and $\sigma-$ are limiting tensile and compressive stresses, respectively. $\mathbb{F} = \{1,2, \ldots, p\}$ is a load case set, where $\alpha$ is the load case identifier, and $p$ represents the total number of load cases.

For the purpose of this work, the layout optimization method is applied via a software tool called PEREGRINE v6.0 [25], a structural layout optimization add-on for Grasshopper. Grasshopper is a visual programming language and environment that runs within the Rhinoceros three-dimensional (3D) CAD application [26].

### 2.4. Post-Processing of CAD and Final Simulation

Layout optimization produces a structurally optimized design; however, design adjustments are necessary to improve the manufacturability of the generated design. Simplifying the design at the manufacturing stage is critical as standard-size beams are used to construct the frame. This can lead to a reduction in overall manufacturing cost and time, especially if a small number of standardized cross-section beams is to be used. The number of selected cross-sections is reduced depending on the application and how often a particular cross-section is used. Sizing the members of the structure is one of the main tasks of this block.

Once the CAD file has been simplified using a few standard beam cross-sections, a final simulation needs to be run to validate the outcomes and sign off the design. For this application, a static structural analysis could give unreliable results; so, a dynamic analysis is required to predict the response of the structure across the loading frequency spectrum of the machining process. In addition, inertial loads from the robot movement are taken into account in this step. Besides the validation of the structural design, simulation results can indicate the sensitivity of the structure and can be used in machining process planning to avoid resonances due to incorrect selection of process parameters.

## 3. Results

### 3.1. Concept and Load Cases

3.1.1. Concept of a Portable Robotic Cell for Hybrid Manufacturing

To maximize the benefits of portability as described in the Introduction section, the envisioned hybrid manufacturing cell must be able to be moved anywhere and put back into operation in a matter of hours. This necessitates the use of a frame that provides a common base between the positioner and the robotic arm, eliminating the need for calibration between them each time the cell is moved. This load-bearing structure of the cell must have sufficient rigidity and high natural frequencies to maintain stability during the machining process, minimizing unwanted vibrations and ensuring high precision. The cell should also be fully enclosed to ensure safety in the working environment. Since the target is to enable hybrid manufacturing, a conceptual design to swap between process heads quickly and automatically has been envisioned, including automatic tool changers as well as toolholders integrated in the process area of the cell.

The portability concept imposes a series of additional limitations for the cell. Its size should be compatible with existing standard dimensions for road transport, while its weight should be kept to a minimum. A three-ton total weight for the cell is pursued, as this corresponds to the upper limit of a lifting category for logistics equipment. Exceeding this weight would directly lead to an increase in the cost and complexity of transport, as specialized equipment would be needed to load/unload the cell and move it into its final position on the shopfloor.

The main purpose of the cell is the repair of large-scale parts, which makes the transport of the cell, rather than the part, more viable. Some indicative parts that the cell can mechanically process are a capacity for up to 500 kg, an approximate height of 1–1.5 m and dimensions of 1 m × 1 m. An indicative cylinder of 0.5 m height and Ø300 mm is shown in the positioner in Figure 2a. The equipment in the cell is therefore selected to meet the aforementioned requirements. The robot model is a Yaskawa multi-purpose handling robot with a payload of 225 kg embedded with its controller. It has a maximum reach of 2702 mm and a repeatability of ±0.05 mm. Yaskawa also supplies the rotary positioner with a payload of 500 kg. The heads for the processes are a Meltio head for the DED process and an HSD spindle for the milling process.

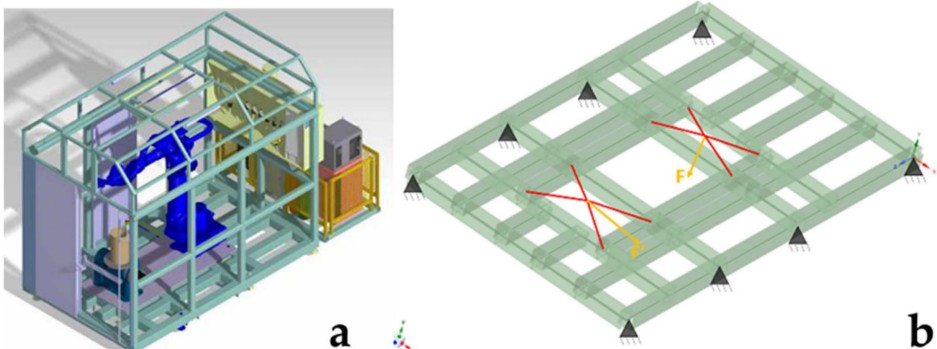

**Figure 2.** (**a**) Portable robotic cell for hybrid manufacturing in isometric view on the left; (**b**) the load-bearing structure design concept with the boundary conditions on the right.

The primary structure must include provisions for the cell enclosure and integrate robust lifting/anchor points for overhead cranes and forklifts. So, lifting and anchor points are integrated in the floor structure of the cell because if they were moved to the top of the enclosure, this would require the enclosure to be load-bearing, increasing the complexity, cost and weight of this solution. Additional height of 0.7 m is added to the standard road transport dimensions which is approximately 2.7 m, which is needed to ensure sufficient working volume for the robot. For this reason, the top of the enclosure is designed to be removable and foldable during transportation. To increase ease of transportation, the cell is separated in two main compartments nested on top of two individual structural platforms: one housing the robotic arm and process heads, which has dimensions of 2.40 × 3.50 m; and one housing the power distribution board, electronics, laser source, inert gas and compressed air supply, which is 2.40 × 1.05 m. As the relative positions of these two components can be changed to adapt to any space, the flexibility of the cell is increased. To facilitate ease of manufacturing, the main structure of the cell is designed as a truss, made up of metallic beams that are welded together.

3.1.2. Load Cases of the Robotic Cell Frame

The robotic cell frame would have to withstand different load cases, both static (during loading, unloading and transportation due to the weight of the individual components), and dynamic (rapid movement of the robotic arm, cutting forces generated by the machining process). This is summarized in the following load cases:

A.   Dynamic analysis for rapid movement of the robotic arm during head change.
B.   Dynamic analysis of the cutting forces during in the different milling poses.
C.   Static analysis for the transport of the robotic cell.

For the first and third load case, a series of simulations were made using a basic conceptual design for the cell structure, including rapid movement between random positions and poses of the robotic arm. It was found that both scenarios resulted in relatively small deformations of the conceptual cell structure.

In addition, high stiffness is crucial only during the machining process. To produce high-quality parts during machining, the structure must be designed with high stiffness, resulting in low deformation under cutting conditions. To ensure an efficient stiffness-to-weight ratio, the target maximum deformation due to machining process forces is set to be close to the minimum machining tolerance limit throughout the frequency range expected. As such, the second load case will be the primary one to be considered in the design of the frame.

The robot's potential poses are simulated in a variety of scenarios. Both the two DoF positioner and robot weights are considered, as are the cutting forces generated during the milling operation. A dynamic analysis of the cell is also carried out, predicting the vibrations generated by the material removal processes. The load cases for the structural part are discussed below.

As there is theoretically no limit to the number of poses the robot can take during a milling process, a minimum of three different cutting poses are simulated. To capture the full working envelope of the robot, the selected poses have a significant variance (Figure 3).

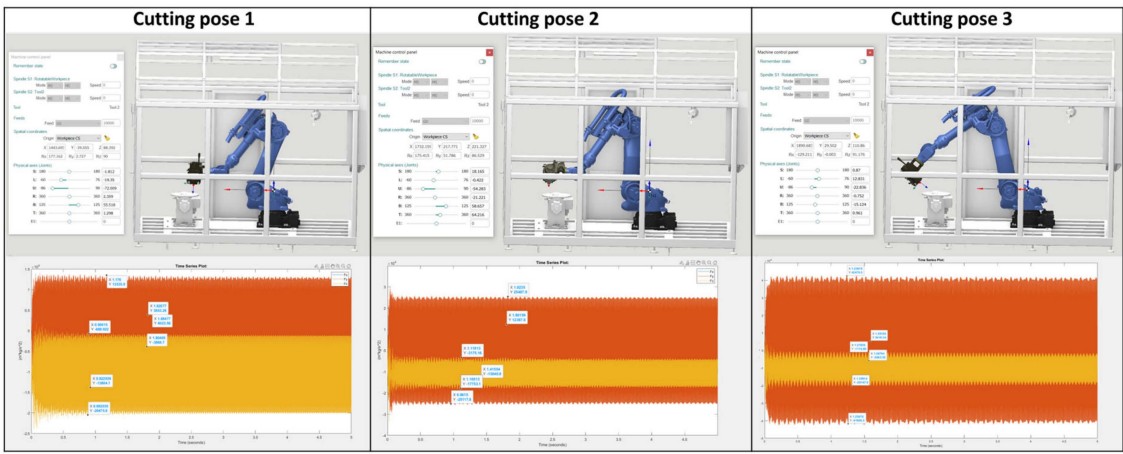

**Figure 3.** Different cutting poses with the corresponding results from MBS.

Since the cell is envisioned to machine components in an HM scenario, it will often have to post-process additively manufactured parts. Hard steels or heat-resistant alloys (HRAs) are the materials used in most AM applications. IN718 is one of the HRAs that results in significant cutting forces and has particularly low machinability. It is also a common material for AM parts. Therefore, this material is used as a worst-case scenario to estimate the cutting forces generated by the process.

In addition, cutting forces calculated using aggressive cutting settings to determine the worst-case scenario are as follows [27], based on the specific force coefficients for this material:

$$F_x = 1650N, \; F_y = 3130N, \; F_z = 230N$$

The results of the aforementioned process across all three poses are summarized in Table 1.

**Table 1.** Cutting forces in XYZ correspond to different poses.

| Forces (N) | Cutting Pose 1 | Cutting Pose 2 | Cutting Pose 3 |
|---|---|---|---|
| Fx | 3860 | 12,710 | 5700 |
| Fy | 13,670 | 25,300 | 42,100 |
| Fz | 9890 | 7290 | 9500 |

### 3.2. Layout Optimization Results

The next step involves the layout optimization that takes place in the Rhino/Grasshopper environment using the PEREGRINE add-on. The architecture of the software includes the creation of blocks and their interconnection. This enables a structured way of programming and simulation and facilitates the identification of errors. Each block is a command with standardized inputs and outputs, making the tool user-friendly.

As the model is set up, several design concepts can be investigated by making small modifications to the blocks of the model and with low computational time. So, three different concepts are presented (Figure 4) with different numbers and types of support. The third concept is selected as it has the lowest volume of all, and a lower number of members compared to the second one. However, some modifications are necessary to establish a connection between the robot and positioner as they cannot be separate parts, and to integrate mounting points for the enclosure.

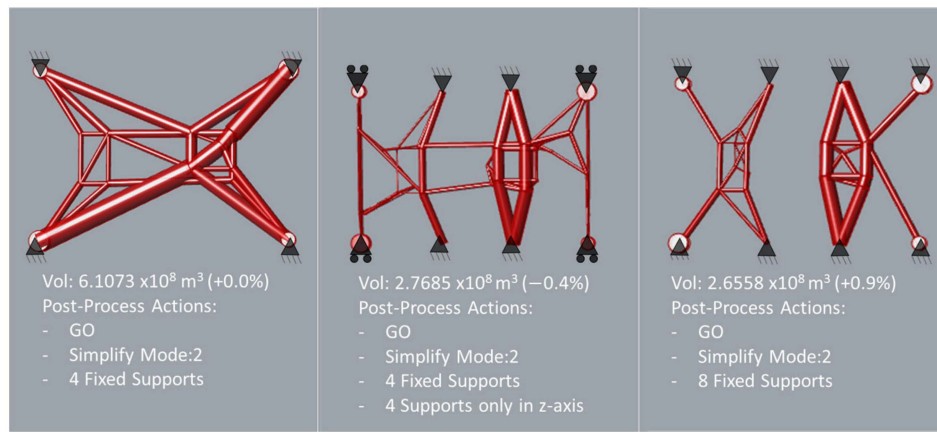

**Figure 4.** Optimization results from PEREGRINE software for three different concepts.

### 3.3. Post-Processing of CAD and Final Simulation Results

This is where the post-processing of the result generated by the layout optimization process takes place. The output file of the layout simulation is in STP format, which allows for post-processing. It is possible to perform a cross-section transformation after measuring the external dimensions of the output STP. Since the design domain in the layout optimization uses a finite height to run the algorithm, the results may include beams in the vertical direction. As the entire platform is subjected to bending loads, beams are merged in this step. To facilitate ease of manufacturing and simplify the bill of materials, three different beam cross-sections (color-coded in Figure 5) are used to construct the complete structure.

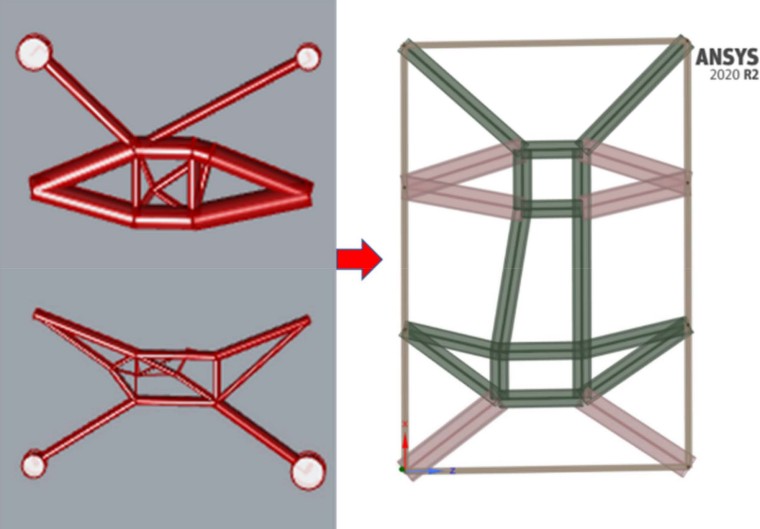

**Figure 5.** Detailed design using the output from the layout optimization as a guideline.

Due to transportation constraints, a hollow rectangular shape was chosen to act as anchor points and maintain a compact platform height. The maximum thickness that can be achieved within the specified dimensions is in the structural beams of the cell. To integrate mounting points for the enclosure, a series of outer beams is added, creating an outer frame (denoted in orange in Figure 5), as the layout optimization does not use any outer beams. Therefore, a beam with small dimensions and the smallest possible thickness is used.

The coordinates of the points are inserted into the ANSYS 2020 R2 environment using the SPACECLAIM add-on and then a beam with the specified cross-section is added using the 'extract' command. After saving, the different sections can be modified. Post-processing facilitates design modification and eliminates the need for beam joint trimming as it is completed automatically. This improves the design for the final analysis stage.

The next step is to validate the detailed design by running an additional dynamic simulation. The results from the second load case are presented in the next section, with all the cutting poses compared in the same diagram. In this section, the results from the dynamic response analysis are presented. First, the tooth passing frequency (*TPF*) is introduced below, where $\Omega$ is the rotating speed of the spindle, and $n$ is the number of cutting edges of the cutting tool.

As the selected spindle has a rotating speed limit of 11,000 rpm and the most commonly used cutting tools have a maximum of six cutting edges, the diagram below is presented for two, four, and six cutting edges.

$$TPF\ (\text{Hz}) = \Omega\ (\text{rpm}) \times n\ \left(\frac{\text{edges}}{\text{rotations}}\right) \times \frac{1}{60}\ \left(\frac{\text{min}}{\text{s}}\right)$$

Figure 6 also shows the spindle revolution ranges that are used for each machining operation. Drilling mainly occurs at the lower frequency range, while milling takes place at the higher frequencies.

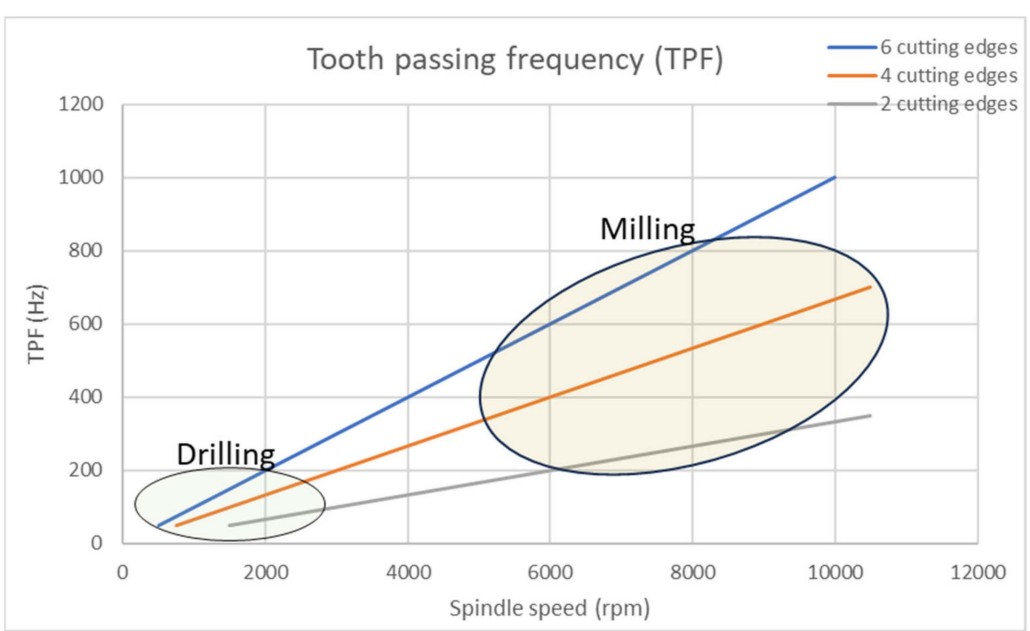

**Figure 6.** Tooth passing frequency for various cutting edges of the tool.

The results of the harmonic response analysis in ANSYS are presented on the following pages for the corresponding load cases in Figures 7–9. For the following results, the *y*-axis of each diagram stands for the displacement, and the *x*-axis for the tooth passing frequency.

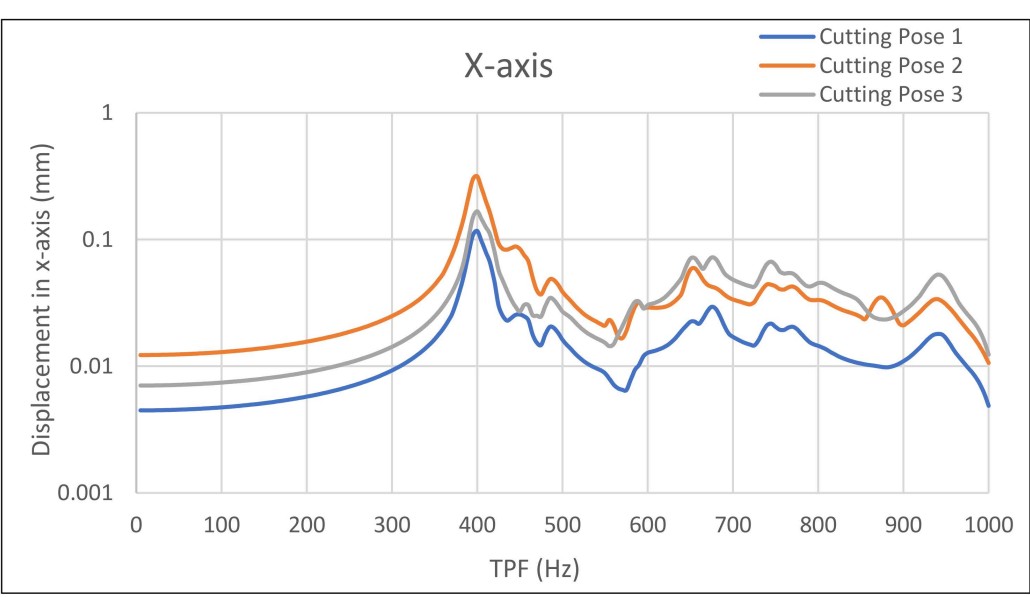

**Figure 7.** Displacement in *x*-axis vs. TPF.

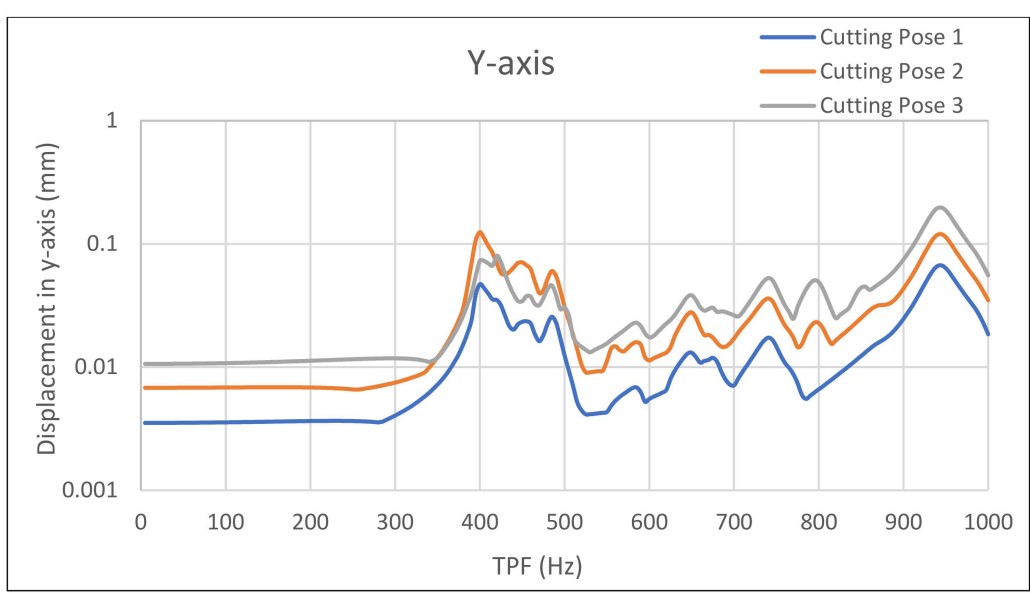

**Figure 8.** Displacement in *y*-axis vs. TPF.

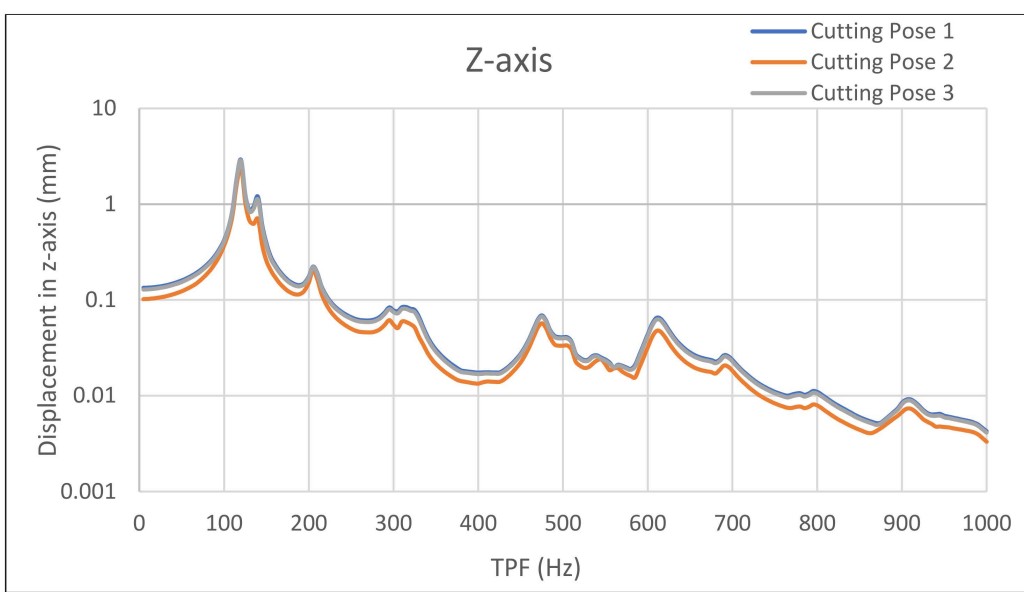

**Figure 9.** Displacement in *z*-axis vs. TPF.

Since it appears that the cutting pose just modifies the amplitude and has no effect on the diagram's quality, the findings are examined for each axis separately. Therefore, the outcomes are similar across the robot's whole working envelope. The *TPF* plot indicates that the primary frequencies for drilling are between 50 and 200 Hz, whereas the main frequencies for milling are between 200 and 800 Hz.

Starting with the *Y*-axis, the optimized design has significantly lower displacements over the entire frequency range, except for the highest band of frequencies (850 Hz), which are insignificant according to the TPF function. Through the optimized design, drilling can be achieved due to minor displacements in this range. This achievement for drilling also applies to the *x*-axis. Also, drilling in the *z*-axis does not affect the process as much as there is no requirement for a tight tolerance in this axis. The optimized design has a displacement of approximately 0.05 mm in the 500–800 frequency range for the *x*- and *z*-axes. This displacement is a critical limit because it is the repeatability of the robotic arm; so, the load-bearing structure does not impair the quality of the parts produced if this limit is not exceeded.

Regarding the manufacturing of the platform, the optimized design is more complex to manufacture than the initial conceptual one (Figure 2b), as beams are required to be cut in the right place and at the right angle to create the nodes where one beam interacts with the other. Finally, the weight of the optimized version has been reduced by 17.6% in comparison with the conceptual design and so the final weight of the cell is lower than the target weight of three tons. This can reduce the operating costs of the cell and make it more attractive for investment.

## 4. Discussion

HM can open new opportunities for repair situations and further flexible manufacturing. Furthermore, the benefits of portable robotics cells are discussed, along with how and where they can affect the supply chain. By using portable robotic repair units, a problem can be solved on-site rather than having to transport the damaged part to a specialized workshop, which has been the standard practice for many years.

The numerous design possibilities of these cells make the suggested method structured and easily modifiable. Optimization might occur at the concept design phase of the robotic cell after a design concept has been selected. The designer can swiftly and effectively verify design concepts with the aid of the layout optimization tool. The result serves as a guide for the phase of detailed design. As the design domain is limited, the possible solutions for layout optimization are limited too. With a larger area, the design challenge is more complex and the difference in the results will be greater. Moreover, the practical boundary requirements precisely address the entire work. The simulation of the milling process was completed using a particular cutting tool and material to provide accurate input for the layout optimization and the final analysis.

On the $y$-axis, layout optimization has a greater impact than on the other axes. This is a result of the side supports. Since the depth of a hole has a lower influence on process quality, the $x$- and $y$-axes have been optimized for frequencies up to 200 Hz, increasing the accuracy of the position of a hole's center. The $y$-axis can alter the milling approach because it behaves better than the other axes. An important benefit of using the $y$-axis as the primary toolpath direction is increased machining precision.

Future research is advised to strengthen the strategy even more. In order to create a more automated framework, the estimation of the process forces can be connected to user needs. The user can estimate milling forces by first choosing the materials, the deposition/machining rate, and the maximum required dimensions of the parts produced. Apart from milling, another conventional process that has been in use is grinding. There is a possibility that the surface roughness will be less than the finishing pass of milling. Certain industries, like the naval, rely heavily on surface roughness for component performance. Thus, a grinding procedure ought to be incorporated into this kind of repair equipment. Further approach validation is another recommendation for the future. When producing this cell, deformation should be measured on the platform of the cell during milling. This will close the design loop for the load-bearing structure by validating the actual deformation.

## 5. Conclusions

HM robotic cells can offer significant benefits. Combining AM and subtractive processes ensures reduced material waste and energy consumption. This will help to meet future sustainable targets for green manufacturing, resource conservation, and the best possible use of materials. Such a cell can also repair existing parts, extending their lifetime, and reducing the need for new ones. In the case of a portable cell, the adaptability of HM enables in situ repair solutions, especially for large-scale parts. The main driver of this work is therefore that this has been designed to be portable, to serve repair applications of large-scale parts through HM. This fact poses limitations in the design of the cell and layout restrictions for ease of transportation, as well as weight constraints.

This work proposes an approach for designing an optimized structure for a robotic cell based on the dynamic loads coming from the machining and the robot. A mechanistic cutting force model is used to estimate the cutting forces based on the real parameters of the process. A previously developed multi-body simulation then analyzes the movement of the robot and predicts the reaction forces at its mount. The transformed cutting forces are used as input to optimize the support structure. This optimization acts as a guide in the detailed design phase. At the end of the approach, the design is validated by a dynamic analysis to check the response of the structure over the full range of spindle speeds. This work resulted in a weight reduction of 17.6% compared to a conceptual design and a maximum displacement of 0.05 mm at the most frequently used machining frequencies.

**Author Contributions:** Conceptualization, P.S. and H.B.; methodology, H.B. and T.S.; software, D.M.; investigation, D.M.; validation, D.M.; cutting force analysis, T.S.; formal analysis, D.M.; writing—original draft preparation, D.M.; writing—review and editing, H.B. and T.S.; supervision, P.S. All authors have read and agreed to the published version of the manuscript.

**Funding:** This research was co-funded by the European Regional Development Fund of the European Union and Greek national funds through the Operational Program Competitiveness, Entrepreneurship, and Innovation, under the call RESEARCH—CREATE—INNOVATE (project code: T2EDK-03896).

**Data Availability Statement:** No new data were created or analyzed in this study. Data sharing is not applicable to this article.

**Conflicts of Interest:** The authors declare no conflicts of interest.

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
