# Peer review of "Process-Driven Layout Optimization of a Portable Hybrid Manufacturing Robotic Cell Structure"

_2673-4117, doi:10.3390/eng5020049_

Round 1

Reviewer 1 Report

Comments and Suggestions for Authors

# Summary of Recommendation

This article evaluates an integrated process to design robotic hybrid manufacturing work cell given material processing (machining) stiffness constraints. The authors propose a process-driven optimization method to balance weight and stiffness of the structure. The method resulted in a reported 0.05mm displacement for 500-800 Hz frequencies in specific directions. 

While the article presents an interesting application of process-driven optimization for a robotic work cell design, the article's title, body of work, and results do not match the research focus. Furthermore, the article suffers from lack of quality in figures, professional writing, and organizational clarity. The article should be refocused on mitigation of cutting forces considering dynamic robot stiffness, and the title should be updated to match. Hybrid (more specifically additive processes) do not play a significant role in this work, as cutting forces do not apply (as acknowledged in the article). Therefore the articles title and focus should center specifically on optimization of a portable robotic machining cell structure, with additive (hybrid) capability as an application. Additionally, the article should be revised to use professional language. For example, "XYZ was done" -> "XYZ was completed". The authors are encouraged to revise for a second iteration, then resubmit the article for review. 

# General Notes

0. Abstract 

-- The abstract must clearly define the type of system. In this case, robotic machining, not hybrid.

-- Include summary of results in the Abstract. Note the 0.05mm displacement and other major numerical achievements. 

# 1. Introduction

-- Include citations for robotic machining cells. 

-- Again, less focus on hybrid. The main contribution is design for stiffness, not subtractive + AM combined. 

-- There is significant literature in the robotic machining space. Plenty of sources to cite.

# 2. Materials and Methods

-- Section 2.1: Cutting force estimating is a field with deep literature. Recommended to find an older / more widely accepted citation for calculating cutting forces. 

-- It is not clear if this method only considers cutting forces, and not robot dynamics. The effect of gravity is mentioned earlier in the text, however, it is not clear if gravity is considered in the simulation.

# 3. Results

-- Adjust images in Figure 3 for consistency. The robot / cell should be viewed from the same perspective for comparison

-- Figure 4, right panel is unclear. Recommend to increase size of the figure, or reduce complexity of the figure such that the information flow is visible.

-- Figure 5, blue text is very difficult to read against a gray background. 

-- Figure 7, use higher quality image (higher DPI)

-- Figure 8, Axis labels are not centered, line markers outweigh the line itself, and the three series plots are not clear. Same for figures 9 and 10. 

# 4.Discussion / 5. Conclusions

-- Statements in the Conclusion section should be moved to the Discussion section. Conclusion should be a concise summary of the work. Additional paths / other notes about the work should be moved to Discussion.

Comments on the Quality of English Language

The article should be revised to use professional language. For example, "XYZ was done" -> "XYZ was completed".

Author Response

Dear Reviewer, 

Please find attached our responses to your comments.

Best Regards,

The Authors

Reviewer 2 Report

Comments and Suggestions for Authors

This is a well written paper that has good technical merit.  The only minor note is that some of the figures are hard to read and are very busy.  The authors may wan to think about how to better present them. 

Author Response

(The authors gave the same response as above.)

Reviewer 3 Report

Comments and Suggestions for Authors

The article presents the results of research related to hybrid production. The authors focused on presenting a production cell enabling the implementation of machinnig processes (such as milling and grinding).

The article proposes a structured approach to optimizing the supporting structure for a portable robotic cell, which is based on the forces generated by production processes. The authors proposed a simulation model that can be used to predict the stiffness of a robot arm.

It should be noted that in the future, robotic mobile stations may provide great support to engineers in the field of equipment renovation (e.g. arc welding, machine processing) as well as the production of spare parts using additive methods.

The approach is very interesting, however, I have small concerns about this work:

1.      Line 95 – the quality of the figure must be improved.

2.      Line 125 – “The low stiffness that robotic arms provide in comparison to their many degrees of freedom is one of their main drawbacks”.

·         In machining applications, robots with high structural stiffness are used (e.g. IRB 2400) - please refer to this.

3.      Line 129 –  “Moreover, robot behavior can be impacted by gravity”.

·         Taking into account gravity forces is done by introducing a special coefficient whose value depends on the method of mounting the robot (floor, wall, ceiling) - please refer to it.

4.      Line 337 – the quality of the figure must be improved.

5.      Lines 366-371 – In the article, the authors mention the reduction of, among others: weight of the work cell. It would be necessary to present the effect of the research and provide numerical or percentage values. In addition, the authors also mention the dimensions of the cell, and here it is also necessary to specify what case was considered. The dimensions and weight of the cell will largely depend on the robot and positioner used. Please refer to a specific example.

6.      The article indicates that the cell should be universal, the authors should indicate the multiplicity of the working space. Refer to the dimensions and weight of objects that can be mechanically processed in the cell.

Comments on the Quality of English Language

Author Response

(The authors gave the same response as above.)

Round 2

Reviewer 1 Report

Comments and Suggestions for Authors

Given the authors insistence on the use of Hybrid technologies as the motivation for this work, and the inclusion of Hybrid technologies in the introductory paragraph, the conclusion of the article should similarly connect back to the motivation of Hybrid. Currently, the conclusion does not mention the use of Hybrid at all, and only focuses on robotic machining. 

Author Response

Dear Reviewer,

Please find attached our response to your comments on our manuscript.

Sincerely,

The Authors
